# Mediating Effect of Brand Image and Satisfaction on Loyalty through Experiential Marketing: A Case Study of a Sugar Heritage Destination

Aileen H. Chen [1,*] and Ryan Y. Wu [2]

1    Department of Leisure, Recreation and Tourism Management, Southern Taiwan University of Science and Technology, Tainan 710301, Taiwan
2    Graduate Institute of Tourism Management, National Kaohsiung University of Hospitality and Tourism, Kaohsiung 812301, Taiwan; ryanwu@mail.nkuht.edu.tw
*    Correspondence: huiling@stust.edu.tw

**Abstract:** While the value of industrial heritage tourism has been recognized, less is known about visitors' experience formation process and its role along with brand image and satisfaction in predicting their loyalty towards industrial heritage destinations. This study adopted the experiential marketing approach to unveil the experience formation process. It also examined the decisive factors of loyalty, by further exploring the mediating effects of brand image and satisfaction on the relationship between experiences and loyalty. A survey questionnaire method was used to collect data from 399 visitors to a sugar heritage destination in Taiwan. The results supported the application of experiential marking in a sugar heritage destination and suggested that visitors' experiences are most driven by how they feel, think, and act. The findings also confirmed the causal relationship and indicated the importance of brand image on loyalty formation and its mediating role between experience and loyalty.

**Keywords:** industrial heritage; experience; brand image; loyalty

## 1. Introduction

Heritage tourism has become a burgeoning area of research, as it was found to help promote economic growth and regional development, and enhance social identity and heritage conservation [1–4]. The term "heritage" is often assigned the role of carrier of historical value of the cultural elements of a society, and as such heritage is seen as a strong attraction for tourists [5]; it may refer to tangible elements including historical buildings, art works and landscapes [6,7] or intangible elements involving the distinctive ways of life and experience of spaces perceived by the visitors as heritage [8]. The latter perspective leads to diversification of heritage and expands to non-traditional areas as industrial production [9], historical theme parks [10], restaurants [11], and seaside resorts [12].

In 2020, there were more than 90 industrial heritage sites inscribed in the World Heritage List [13]. With the advancement of new technology and globalization, many industrial facilities are abandoned across countries. Some of these sites have been rejuvenated and transformed into travel destinations in an attempt to celebrate history and at the same time reshape city image. Some examples of this include Big Pit at Blaenavon in South Wales (former coal mine, now part of Blaenavon World Heritage Site), The Ruhr region of Germany (once one of Europe's largest coal mining areas), and Jiufen in Taiwan (former gold mine known from the movie *City of Sadness*). These areas successfully integrate their valuable heritage into tourism efforts and attract high numbers of visitors [14].

Sugar production, once the major industry in Taiwan, is now facing severe price wars and is struggling to survive. There were 42 sugar factories in 1945, and now only two of them remain in operation [15]. Those abandoned factories stayed idle until the 2000s when industrial tourism gained much attention from the world. Many facilities involving sugar

production were renovated and promoted to be tourism attractions, including warehouses, sugar trains, production equipment, etc. Tourism-oriented development was deemed a viable strategy to activate those heritage sites and make visitors aware of the sugar production process, local culture, and traditions. However, as one segment of industrial heritage tourism, sugar heritage destinations have to compete with many other places promoting themselves using similar mechanisms, namely presenting production processes via interpretation. In other words, the destination needs to be unique and identifiable in its offerings to be selected as a final destination [16]. In this sense, the concept of destination branding is critical to create consistent identity and uniqueness differentiating a destination from other places. Through a set of marketing activities, a destination is able to be identified and differentiated from others by cultural attractions, heritage, landscape, hallmark events, and memorable travel experiences. All of these discerning characteristics collectively create a positive destination brand image, a strong emotional connection with visitors, and can further influence visitors' destination choice and intention to recommend [17].

Loyalty has been widely examined in the context of cultural tourism. A review of loyalty studies shows that satisfaction, image, emotions, quality/performance, and motivation are important influential factors [18–21]. As implied by Kim, Wong, Chang, and Park [22], the selecting of loyalty factors should depend primarily on the research nature and contexts since tourists' loyalty might differ by industry and market stage of product life cycle. Many studies have documented a path relationship between experience, satisfaction, and loyalty [23–27]. Results of this body of research have shown that tourists' experiences directly influence their evaluation of the travel process, and hence determine their satisfaction and subsequent behaviors. The influence of brand image on satisfaction and loyalty have been examined in consumer behavior literature [28,29], and satisfaction might fully mediate the influence of brand image on loyalty in the tourism field [30]. In the era of the experience economy, experience has been viewed as the core for heritage tourism and is significant in the building of destination brand image [31,32]. Therefore, this study sought to delve deeper and expand the knowledge base associated with experience, brand image, satisfaction, and loyalty in the context of heritage tourism.

In recent years, Taiwan has made many efforts to promote cultural heritage development, and sixty-seven out of 188 sugar attractions were listed as national cultural heritage assets (*TaiSugar Communication Monthly*, October 2016) [33]. The International Committee for the Conservation of the Industrial Heritage (TICCIH) under the UN even declared one of the sugar factories in Taiwan a demonstration site. Sugar heritage has become the representative of sustainable tourism development in Taiwan. However, after a review of industrial heritage tourism studies, we have found that most of the related research focused on the issues of planning, impact, and management (see Table 1) while ignoring the core issues, namely visitor experiences and the formation process. Sugar heritage tourism is a new research area which has received less attention from researchers. Insufficient research has been done on the visitors to sugar heritage attractions, and specifically, their perceived experiences and how these influence subsequent behaviors.

**Table 1.** Brief summary of studies on industrial heritage tourism.

| Authors | Study |
|---|---|
| Balcar and Pearce [34] | Differences of site characteristics, development, management and visitor profiles among eight gold/coal mining heritage sites on the West Coast of New Zealand. |
| Bramwell and Rawding [35] | Examine the rationales behind image reshaping in five industrial English cities. |
| Edwards and Llurdes [6] | Examine the potential of industrial heritage, specifically mining areas, transforming into tourist attractions. A typological framework of industrial heritage sites was developed. |

**Table 1.** *Cont.*

| Authors | Study |
| --- | --- |
| McBoyle [36] | Adopt a green image strategy in industrial tourism attractions to enhance firms' environmental reputation by encouraging self-directed improvement in line with environmental accreditation schemes. |
| Harris and Masberg [37] | A cross case analysis on the vintage trolley operations identifying 12 factors to success. |
| Kerstetter, Confer, and Bricker [38] | Examine the relationship between tourists' visitation types and patterns. |
| Rudd and Davis [39] | Develop industrial heritage tourism to generate a positive image for the copper industry in Utah and to alleviate public fears about pollution and environmental degradation. |
| Prentice et al. [10] | Experiences and benefits segmentation of visitors to a mining-themed industrial heritage park in South Wales UK. |
| Baum [40] | Examine the policy issues for diversifying fishery industry in two island communities: Iceland and Newfoundland. |
| Caffyn and Lutz [41] | The policy issues designed to encourage a community focus in urban industrial heritage tourism. |
| McIntosh and Prentice [42] | The experiential and emotive processes in visitors' interaction with industrial heritage attraction settings to affirm authenticity. |
| Wanhill [43] | The key issues involved in setting up mines and industrial remains as tourist attractions in South Wales UK. |
| Pretes [44] | Examine tourist–local interactions and the use of tourism as a vehicle for narrating an indigenous discourse at the silver mines in Potosí, Bolivia. |
| Cole [45] | Using economic, social, and environmental perspectives to examine the development of mining heritage tourism towards sustainable objectives. |
| Billington [46] | Discuss how federal investment attracts private investment in the Blackstone Valley, America's industrial birthplace, and why people are returning. |
| Frost [47] | Examine the relationships between heritage tourism and shopping in two towns (once gold mines): Castlemaine and Maldon, Australia. |
| Xie [9] | Evaluate the proposal for a Jeep Museum in Ohio from six key attributes of developing industrial heritage tourism. |
| Ruiz Ballesteros and Herna'ndez Ramı'rez [48] | The relationship between local community (namely community identity) and the development of mining heritage tourism in southern Spain. |
| Vargas-Sánches, Plaza-Mejía, and Porras-Bueno [49] | The attitudes of local residents towards tourism development in Minas de Riotinto (traditionally involved in mining activity) and the relationship to community satisfaction. |
| Buultjens, Brereton, Memmott, Reser, Thomson, and O'Rourke [50] | Indigenous involvement in tourism in the Weipa region of northwest Queensland and the role of a mining operation called Rio Tinto Aluminium in assisting this development. The facilitator approach adopted is likely to limit the effectiveness of the mine's efforts. |
| Donohoe [51] | Propose a sustainable marketing model and test it on a World Heritage Site—the Rideau Canal in Canada. |
| Kastenholz, Carneiro, Marques, and Lima [52] | Examine the experience of tourists and residents in a small village in Central Portugal that uses its heritage and traditions to promote rural tourism. |
| Spencer and Nsiah [53] | A case study on the economic consequences of community support for tourism development in a heritage fish hatchery. |
| Ma, Weng, and Yu [54] | A case study on the structural evolution of a historic water town in China, and the role of scale economies and market size on it. |

**Table 1.** *Cont.*

| Authors | Study |
| --- | --- |
| Wu, Xie, and Tsai [55] | Examine the key attributes of attractiveness for salt destinations in Taiwan and the preferred experiences and programs. |
| Xie [56] | Propose a life cycle model of industrial heritage development and apply it to the LX Factory in Lisbon, Portugal. |
| Lin [57] | Explore visitors' perceptions of authenticity and its impact on intrinsic value in Pingxi, Taiwan. |
| Goulding, Saren, and Pressey [58] | Investigate the commercial staging of history in a case of an industrial museum and examine how the past is experienced and understood through a series of factors. |
| Lin and Liu [59] | Explore the construct of authenticity in the context of industrial heritage in Taiwan and examine its relationship with motivation and loyalty. |
| Peng and Tzeng [60] | Propose a hybrid-modified MADM model to develop performance-improving strategies for industrial heritage tourism. |
| Su, Dong, Wall, and Sun [61] | Conceptualize a multiple value system of agricultural heritage in China to explore the approach of integrating the heritage values with tourism. |
| Yan, Shen, Ye, and Zhou [62] | Assess the effects of awe experience, authenticity experience, and perceived value on visitors' behavioral intentions towards the industrial heritage park in east China. |
| Marques, Fazito, and Cunha [63] | Explores the conflicts between tourism development and mining, and concludes that more balanced human development should be ensured in public planning processes. |

This current study was therefore conducted by using an experiential marketing approach to explore the various experiences visitors perceive during the visit. In order to investigate the roles of experience, brand image, and satisfaction in predicting loyalty, data from visitors to a sugar heritage attraction were used. The mediating effects of brand image and satisfaction were further identified to facilitate a more complete understanding about the formation of subsequent behaviors. As a result, the contribution of this paper is threefold. First, we explicitly examine the relationships between experiences, brand image, satisfaction, and loyalty. It is important to confirm the applicability of experiential marketing approaches in a destination context and establish an influential path depicting the relative influence of experiences, brand image, and satisfaction on loyalty. The less researched relationship of experience to brand image is sought to be established and confirmed. Second, to advance our understanding about the decisive factors of loyalty, the mediating effects of brand image and satisfaction on the relationship between experiences and loyalty are further evaluated, given inconclusive evidence in both the tourism and marketing literature as stated above. Third, this research contributes to the limited body of knowledge addressing tourists' experiences while visiting a sugar heritage destination.

## 2. Literature Review

### 2.1. Industrial Heritage Tourism and Experience

Most researchers agree that heritage tourism is one of the most significant and fast growing segments of tourism [64]. Among the segments, industrial heritage tourism represents a promising niche market that assists in regional restructuring and economic development. Industrial heritage tourism has been defined as "the development of touristic activities and industries on man-made sites, buildings and landscapes that originated with industrial processes of earlier periods" [6] (p. 342). Themes of industrial heritage are classified into ten categories: extractive industries (e.g., iron, gold), bulk product industries (e.g., textiles), manufacturing industries (e.g., machine manufacture), utilities (e.g., water supply), power sources and prime movers (e.g., windmills), transportation (e.g., railroads),

communication (e.g., telephones), bridges, trestles or aqueducts (e.g., movable bridges), building technology (e.g., roof systems), and specialized structures/objects (e.g., dams) [65]. Of these, many sites have developed tourism, with mines, metal working sites, factories, and transportation systems being the most visited.

Industrial heritage has been called the "landscapes of nostalgia" [66] (p. 566). It is "not just about the monuments and artifacts that remain, important though they are, but also about the people and communities whose lives, enterprise and energy have made the areas what they are" (the European Route of Industrial Heritage, adapted from Cole [45] (p. 481)). Visits to these industrial heritage sites give tourists nostalgic and novel experiences about traditional manufacturing and processing systems. McIntosh and Prentice [42] examined the process of visitors interacting with an industrial heritage site and noted that "insight" from the interaction process represented a key component of the experiences visitors reported. Heritage settings inducing visitors' personal, familiar, or affective responses were appreciated the most. In other words, visitors went to these heritage destinations to look for unique experiences that were personally meaningful and derived from the interaction between the staged activities and the individuals. In this regard, visitors become an actor rather than spectator [67]. To involve the visitor as a participant or a co-producer, heritage settings must take into account marketing activity offerings and visitor perceptions. Determining which marketing activities would induce deeper and more positive experiences from visitors appears to be a key issue and requires further examination.

The main idea of experiential marketing is identifying the core of products or service offerings and then connecting these to intangible, tangible, and interactive experiences that enhance the perceived value and help customers make their purchasing decisions [68]. In other words, experiential marketing focuses more on customers' sense and emotions' stimulation than on the functions of product/service offerings. After integrating the different aspects of experience formation, Schmitt [69] further proposed five types of consumer experiences that can be created by marketing stimuli, including sense, feel, think, act, and relate, which are termed strategic experiential modules (SEMs). The sensory experience includes esthetic pleasure, excitement, and satisfaction that can be derived from visual, sound, smell, taste, and touch stimuli. The affective experience appeals to customers' feeling and emotions that range from mildly positive moods to intense emotions of joy or nostalgia. The think experience includes convergent and divergent thinking induced by surprise and provocation. The act experience involves encouraging changes in lifestyles, interaction, and behaviors. Finally, the relate experience expands beyond the individual's aspect and relates an individual to a broader social system (e.g., a subculture). Among these five experiences, sense was considered to be powerful for brand building [70] and easier to start with, while affect was the experience most influential on subsequent behaviors when it occurred during the interaction.

Since the introduction of the experiential marketing concept and SEMs by Schmitt [69], there have been limited research articles on related issues. Many of the tourism studies conducted concentrate on the application of experiential marketing [71,72] and scale development [73]. Research using the experiential approach in the context of industrial heritage tourism is even less common. A brief review of industrial heritage tourism research published in the following 10 major tourism journals—Annals of Tourism Research, International Journal of Tourism Research, Journal of Travel Research, Journal of Hospitality and Tourism Research, Journal of Travel and Tourism Marketing, Journal of Destination Marketing and Management, Current Issues in Tourism, Journal of Sustainable Tourism, Tourism Management, and Tourism Management Perspectives—is outlined in Table 1. The review indicated that existing studies have mostly concentrated upon four areas: the planning and management issues of turning industrial heritage sites into tourist attractions, its relationship to community, image reshaping for industrial cities, and visitor experiences and authenticity. Clearly, planning-related topics, community support, and the

mining industry predominate in the research field, while few studies on visitor experiences were undertaken.

Experiences are private events that are perceived and often generated from observation or direct participation in marketing activities as provided before or after purchase [69]. As a marketer, providing the right stimuli (marketing efforts) prompting desired visitor experiences is important. Therefore, in this current study, we measured visitors' perceived experiences after undergoing the marketing efforts (experiential marketing) in an industrial heritage setting.

### 2.2. Brand Image

Brand image is considered as the entirety of impressions that consumers receive from many sources [74]. These sources could be from media reports, word-of-mouth, personal experiences, or media advertising. Later, Keller [75] defined brand image as "consumer perceptions of a brand as reflected by the brand associations held in consumer memory" (p. 3). These associations are the informational links stored in the minds of the consumer and can be reinforced as well as evoked when exposed to marketing stimuli. Kotler and Armstrong [76] indicated brand image as the set of beliefs about a particular brand that help differentiate it from other brands. These beliefs are perceived based on brand attributes and will differ depending on personal experiences, selective memory, and encoding. Brand image plays an important role in the consumer's decision process when evaluating alternative brands, since it signifies higher quality and less purchasing risk that leads to higher purchasing intention [77,78].

As for dimensions of brand image, Park, Jaworski, and Maclnnis [79] developed a framework, called brand concept management (BCM), to facilitate brand image communication and higher market performance. In the framework, the brand concept is the core which guides us through the sequential process of image building, and can be classified into three types depending on customer needs. These include: (1) functional concept: emphasize the brand's performance in solving consumption-related problems (e.g., destination providing entertainment as well as educational programs); (2) symbolic concept: emphasizes the brand's heightening in self-identification, self-expression, and group membership (e.g., destination linked to locations at film programs); and (3) experiential concept: emphasizes the brand's sensory satisfaction and cognitive stimulation, and especially highlights the experiential aspects of fantasy, fun, and pleasure associated with the consumption process (e.g., destination with a variety of fun activities). These three aspects of brand image mirror the three type benefits of brand associations by Keller [75] and are also suggested by many researchers [80–82]. Given that the definition by Keller [75] has received the most support [83–85], therefore, in this study we modify the definition by Keller and adopt the classification by Park et al. [79] to define brand image as "visitors' perceptions of a destination brand as reflected by the brand associations held in memory, and that consists of three components: functional, symbolic and experiential image".

### 2.3. Brand Image and Experience

In the model of customer-based brand equity, Keller [75] used an associative network memory model to explain the process of image forming: the memory is composed of a set of nodes and interconnecting links where nodes signify stored concepts or information while links represent the strength of association between nodes. For a specific association to appear, the nodes containing specific information must be activated first and the other nodes with desired concepts must be strongly connected to the activated nodes. The process starts with a stimulus from the external environment (usually directed by marketers, such as a nostalgic atmosphere in destination), nodes containing information related to that atmosphere being activated (e.g., evokes the scene from childhood), and then those favored nodes with strong links to the activated nodes being activated (e.g., remembers related events and story in that scene). Cai [83] also stressed that brand image is formed through a process that starts with selecting one or more brand elements and then establishing brand

associations that reflect the attributes, affective, and attitudes components of an image. In other words, by carefully selecting appropriate marketing stimuli, marketers could induce and reinforce visitors' association to a brand, which could lead to a positive brand image. These stimuli could be in the forms of sensory, affective, intellectual, behavioral, and social stimuli.

The relationship between experience and brand image has been proposed and verified by many researchers. For example, Padgett and Allen [54] discussed the potential to communicate brand image through a narrative approach and recommended the use of both narrative and argumentative advertisements to convey a functional and experiential brand image. Berry [31] stressed that the actual experience visitors perceive is more powerful than any advertisement to brand image building. Ghaffari, Abasi, and Monfared [86] showed that all dimensions of cultural tourists' experience have influence on destination brand image. All of these papers indicated the importance of experience to brand image. As a result, we posit:

**Hypothesis 1.** *Visitors' experience has a positive effect on their brand image towards the destination.*

### 2.4. Experience, Satisfaction, and Loyalty

Mehrabian and Russell [87] proposed that environmental stimuli influence an individual's affective states, which in turn influence approach or avoidance responses. Following the same direction, Schmitt [41] introduced the construct of experiential marketing and argued that customers not only want functional features and benefits but the offerings which touch their hearts, stun their senses, and stimulate their thinking. He further recommended five types of experience-inducers for managers to create the desired experiences. Marketers can engage customers in the consumption process by using services and goods in a way that creates memorable events, and which further lead to revisit intentions and positive word-of-mouth. Phillips and Baumgartner [88] explained the satisfaction formation process from an experiential perspective and stressed that the satisfaction responses can be described as an experientially motivated process in which consumers approach consumption, form expectations (based on the experiential outcomes), experience (exposing to marketing efforts), and evaluate. As such, it is reasonable to assume that an individual with positive experiences is more likely to be satisfied with the consumption and reveal more positive loyalty behaviors.

In consumer research, many studies are conducted to examine the influence of different environmental stimuli on satisfaction. These environmental stimuli include color, scent, lighting, background music, and cleanliness [89,90] and have been proved to enhance customers' satisfaction and purchasing behaviors. In tourism areas, some researchers have empirically examined the relationships of experiences to satisfaction and loyalty. For example, Barnes et al. [73] and Milman and Tasci [91] indicated that affective (feel) experiences can directly influence satisfaction and loyalty behaviors. Stavrianea and Kamenidou [27] and Chen, Wang, Li, Wei, and Yuan [18] confirmed the direct influence of tourist experience on satisfaction and loyalty. Thus, we posit:

**Hypothesis 2.** *Visitors' experience has a positive effect on their satisfaction towards the destination.*

**Hypothesis 3.** *Visitors' experience has a positive effect on their loyalty towards the destination.*

### 2.5. Brand Image, Satisfaction, and Loyalty

The evaluation of experience is difficult given the intangible and high-risk nature of tourism, and hence previous visits play an important role in forming visitors' satisfaction. Visitors need to collect information internally and externally to form their own beliefs and judgments towards a destination. The information obtained internally involves the activation of knowledge stored in the memory [92]. In the context of a destination brand, the knowledge could be those attributes related to a specific destination. When visitors

perceive certain destination attributes, they will generate expectations in satisfying specific needs which can further elicit satisfaction and encourage future loyalty behaviors [93]. Kotler, Bowen, and Makens [94] suggested that customers tend to have a positive experience towards a purchase when they had a good image of that product before obtaining it. Taking this principle to tourism areas, visitors having a favorable image towards a destination would perceive their onsite experiences positively, and that leads to a higher level of satisfaction and loyalty.

In addition, it is argued that image measurement cannot be based on attributes alone but on the value and benefits from using the brand [95]. This coincides with Keller's [75] model that stressed the favorability of brand image as a function of needs fulfillment benefits and attributes, and also implies the importance of benefits-based image on consumers' satisfaction. The influences of brand image on satisfaction and loyalty have been widely examined in consumer behavior literature [29,96], and further verified in the field of tourism [27,30]. Sondoh et al. [28] investigated the influence of brand image on satisfaction and loyalty, and indicated that the influences vary depending on the brand image components. For example, the image components of functional and appearance enhancement showed positive influences on loyalty but the components of symbolic and experiential images did not. Thus, we posit:

**Hypothesis 4.** *Visitors' brand image has a positive effect on their satisfaction towards the destination.*

**Hypothesis 5.** *Visitors' brand image has a positive effect on their loyalty towards the destination.*

### 2.6. The Mediating Role of Brand Image and Satisfaction

Identifying the determinants of loyalty will allow managers to concentrate on the major influential factors that lead to customers' repeat purchases or visits. Previous studies have shown that satisfaction, quality/performance, image, emotions, and motivation are good predictors of loyalty [20,21,97], with satisfaction displaying a strong relationship to loyalty [98,99]. Satisfaction typically plays a mediating role between visitors' experiences and loyalty. For example, Chao [100] reported that brand image and satisfaction play very important roles as mediators to establish customer loyalty. Similarly, Han, Back, and Barrett [101] found that satisfaction fully mediated the relationship between affective experiences and loyalty behaviors. However, this role is not consistent across all study results. Barnes et al. [73] found that satisfaction only partially mediated the relationship of experience and loyalty behaviors, while Prayag et al. [25] even reported a nonsignificant mediating effect of satisfaction. The same situation applies to the effect of brand image. In a study of airline customers, Brodie et al. [96] found that brand image has no direct effect on loyalty, but affects loyalty through its influence on customer value. Faullant et al. [30] further noted that the importance of image and satisfaction on predicting loyalty differs depending on visitors' experiences, with image more influential for repeat customers. Still, in the tourism field, research investigating the mediating roles of brand image and satisfaction on the relationship between experience and loyalty is limited. It is argued that satisfaction is a necessary but insufficient condition for encouraging positive loyalty behaviors [102]. In other words, loyalty cannot be predicted by satisfaction alone and it is best explained by combining both satisfaction and other variables such as brand image and affect (experiences) [72,103–105]. Based on the above discussions, combined with hypothesis 1 (the influence of experience on brand image), the following hypotheses are posited:

**Hypothesis 6.** *Visitors' satisfaction has a positive effect on their loyalty towards the destination.*

**Hypothesis 7.** *Visitors' brand image mediates the effects of experience on their loyalty towards the destination.*

**Hypothesis 8.** *Visitors' satisfaction mediates the effects of experience on their loyalty towards the destination.*

As described above, visitor experience may affect their loyalty, and the influence might be mediated by satisfaction and brand image. These relationships have been reported in the context of cultural tourism, and were proposed to be applicable to the heritage tourism sector, which is a segment of cultural tourism. Hence, a conceptual model (Figure 1) is proposed in the current study to test the hypothesized relationships in the context of sugar heritage tourism.

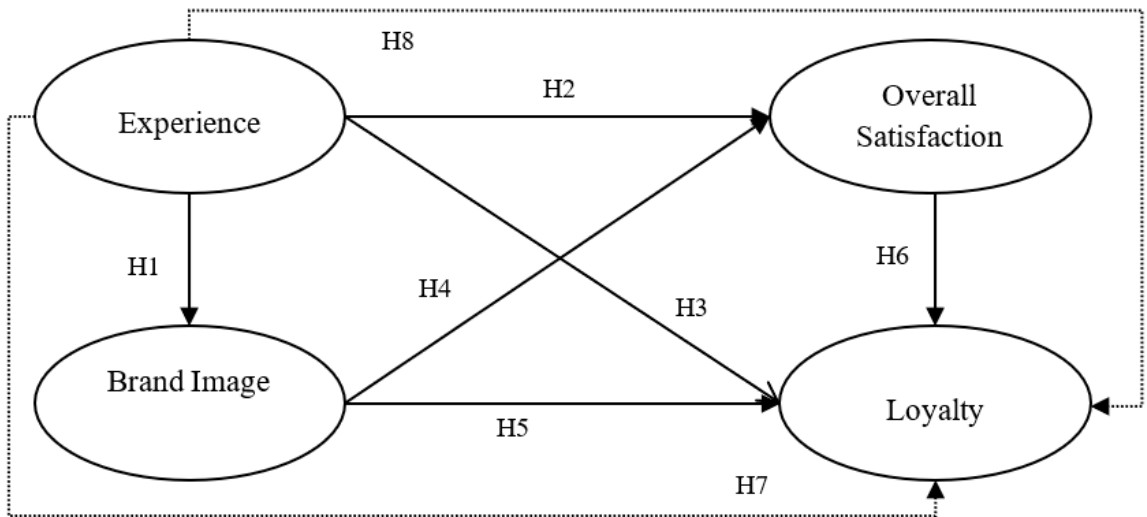

**Figure 1.** The proposed structural model.

## 3. Research Design and Methods

### 3.1. Study Sites

We chose Ten Drum Culture Village, a sugar heritage attraction renowned for the performance of the Ten Drum Art Percussion Group, as the study site. Ten Drum Culture Village is located on the outskirts of Tainan, Taiwan, which was once a sugar refinery and stayed idle for years. After the re-planning and renovation by the founder of the Ten Drum Art Percussion Group, this village recreates the past beauty of sugar refineries via the old buildings, equipment, and architecture. Featuring innovative yet authentic Taiwanese spirit drum music, it also became the first international drum music themed arts village in Asia. A variety of attractions are on offer, including a drum museum which was transformed from a warehouse, the drum education area where visitors can join in with the performers, the molasses tanks which were once used for molasses storage and now are used as museums and a coffee shop, as well as sky walks connecting these three tanks. There is also the water tank theater, which was formerly a water tank to cool hot water discharged by sugar refining, and was rebuilt as an outdoor theater for 1500 people, the sugar refinery chimney, which was part of the Taiwan sugar industry from the 19th century, various recreational facilities, and most importantly, the live drumming performance which integrates various instruments (e.g., flutes) other than drums, martial arts moves, lighting effects, and dry ice to induce visual and audio excitement. Facing the challenge of continuously attracting visitors and audiences, Ten Drum Culture Village exercises various marketing strategies through the development cycle from performance marketing and teaching marketing, events marketing, performance marketing at a fixed site, to strategic alliances with other industries (i.e., movie). It attracted more than 470,000 visitors in 2017, and received about 380,000 visitors annually in the years of 2018 and 2019 [106]. The shift from arts-oriented to customer-oriented marketing has made it become one of the most popular destinations in Tainan with a good word-of-mouth reputation.

### 3.2. Data Collection

The study population consisted of tourists who attended the guided tour in the village and were at least 18 years old. We used an on-site questionnaire survey to collect data. The convenience sampling approach was used in this study. A group of interviewers were trained to participate in the process of collecting data to ensure the validity of questionnaire responses. These trained interviewers were stationed at sites which are the last stops for visitors participating in the guided tours in the village. The questionnaires were handed out to visitors who were willing to take part in the survey after briefly explaining the research purpose. As a result, a total of 499 completed responses were obtained.

### 3.3. Survey Measures

A preliminary list of measures was prepared through the related literature review and is as follows.

To assess visitors' experiences at Ten Drum Culture Village, five types of experiences (sense, feel, think, action, and relate) were adapted from the strategic experiential modules (SEMs) of Schmitt [69], serving as the measurement constructs, with 23 items generated from works of Brakus, Schmitt, and Zarantonello [107]. In their study of brand experience, Brakus et al. chose items from an extensive literature review on the proposed five dimensions and identified the dimensions across different products and services using factor analysis. After a series of five studies, they concluded with a four-factor structure of brand experience, including sensory, affective, behavioral, and intellectual dimensions. Given the fact that their study aimed to measure more general brand experiences instead of one specific setting, we employed the original five dimensions in our study. After a pilot study, one item was removed based on the results of item analysis and reliability analysis. The finalized measurement of experiential marketing consisted of 22 items.

Brand image of the Ten Drum Culture Village was measured using 13 items adapted from the work of Park et al. [79]. These items were classified into three dimensions of functional, symbolic, and experiential. After the pilot study, all items were obtained. Satisfaction with the trip was measured by a single-item ten-point rating scale, as in some previous research studies [108,109], by asking the respondents: 'How would you describe your experience in this village?' Respondents were asked to rate their experience on a ten-point scale (1 = very unsatisfactory to 10 = very satisfactory).

Loyalty to Ten Drum Culture Village was measured using two main constructs from Bigné et al. [109]: the intention to revisit and to recommend. A five-point scale where 1 = least likely and 5 = most likely was used. For all items measuring experiences and brand image, respondents were asked to indicate their levels of agreement with a five-point Likert-type scale, where 1 = strongly disagree and 5 = strongly agree. Eight sociodemographic questions were also included in this questionnaire.

### 3.4. Data Analysis

We used SPSS v25 to conduct descriptive analysis and item analysis, and then structural equation models (SEM) using AMOS v24 in the model testing and hypothesis testing. Prior to beginning statistical analysis, the data were screened for outliers by examining $z$ scores and missing values to assure more accurate results from the statistical tests. Three steps were followed to test the hypotheses. First, a two-step confirmatory factor analysis approach was utilized to evaluate each construct separately, followed by the validation of the overall measurement model [110]. Second, examination of reliability, convergent validity, and discriminant validity were conducted to ensure the models' psychometric properties. Third, the proposed structural path model with two mediators was estimated to test hypotheses. In this study, we used bootstrapping in AMOS to test the mediation effects instead of the causal steps approach suggested by Baron and Kenny [111]. Although the procedures by Baron and Kenny is the most frequently used, bootstrapping is more powerful than the causal steps approach to testing multi-mediator models, as suggested by MacKinnon, Lockwood, and Williams [112]. Further, we used the PRODCLIN2 program

by MacKinnon, Fritz, Williams, and Lockwood [113] to determine the existence of specific indirect effects and conducted approaches by Preacher and Hayes [114] to identify the relative importance of these indirect effects. In doing so, we determined to what extent specific variables mediated the effects without compromising with increased bias.

## 4. Results

### *4.1. Profile of Sample and Responses*

As reported in Table 2, female visitors outnumbered male visitors (57.7% vs. 42.3%). Regarding the age distribution, most visitors (57.6%) fell in to the age range of 26 to 45, followed by the 46 and above group (29%). They were well educated, with nearly 68% of them having a college degree or above. The majority of respondents worked in service industry (68.3%) and reported an income level between NT 30,000 and 49,999 (39.7%), followed by less than NT 30,000 (32.9%), which might be attributed to the unpaid occupation of students, housekeepers, and the retired. In terms of residence, the largest group, 48% of the sample, came from the southern part of Taiwan, followed by the northern part (35.1%). Seventy-two percent of visitors were traveling as a group, and travel agencies were the major sources to obtain information. This statistic is consistent with the marketing efforts adopted by Ten Drum Culture Village, which cooperates with travel agencies to attract group travel. The results of descriptive statistics (Table 3) showed that visitors tend to rate highly each dimension of experience, with all item means being close to or larger than 4. They also revealed a relatively favorable brand image and were satisfied with the overall experiential environment (*M* = 8.33). As result, the respondents showed a high level of loyalty to the destination, with all item means being larger than 4.

**Table 2.** Sample profile (*n* = 499).

|  | Frequency | % |  | Frequency | % |
|---|---|---|---|---|---|
| Gender |  |  | Monthly income (NT) households |  |  |
|   Male | 211 | 42.3 |   <30,000 | 164 | 32.9 |
|   Female | 288 | 57.7 |   30,000~49,999 | 198 | 39.7 |
| Age |  |  |   50,000~69,999 | 88 | 17.6 |
|   18–25 | 75 | 15.0 |   >70,000 | 49 | 9.8 |
|   26–35 | 155 | 31.1 | Residence |  |  |
|   36–45 | 132 | 26.5 |   Tainan City | 85 | 17.0 |
|   46–55 | 90 | 18.0 |   Northern area | 175 | 35.1 |
|   56 and older | 53 | 10.6 |   Central area | 85 | 17.0 |
| Education |  |  |   Southern area | 154 | 30.9 |
|   Secondary school or below | 33 | 6.6 | Travel style |  |  |
|   High/vocational school | 125 | 25.1 |   Individual travel | 141 | 28.3 |
|   University/college | 300 | 60.1 |   Group travel | 358 | 71.7 |
|   Graduate school or above | 41 | 8.2 | Sources |  |  |
| Occupation |  |  |   TV | 50 | 8.5 |
|   Government/education agencies | 32 | 6.4 |   Newspapers | 30 | 5.1 |
|   Manufacturing industry | 15 | 3.0 |   Websites | 82 | 13.9 |
|   Service industry | 338 | 67.7 |   Brochures/DM | 40 | 6.8 |
|   Technician | 30 | 6.0 |   Families/friends | 108 | 18.3 |
|   Unemployed (students, retired) | 84 | 16.8 |   Travel agencies | 281 | 47.6 |

**Table 3.** Results of the measurement model for each construct.

| Constructs and Items | Mean | Factor Loadings | *t*-Value | SMC | CR | AVE |
|---|---|---|---|---|---|---|
| **Experience** | | | | | | |
| Sense | | | | | 0.74 | 0.49 |
| The landscaping in this village attracts my visual attention | 3.61 | 0.668 | NA | 0.446 | | |
| The trip in this village is interesting | 3.86 | 0.652 | 12.486 | 0.425 | | |
| This village is full of sensory appeal | 4.03 | 0.772 | 14.099 | 0.596 | | |
| Feel | | | | | 0.78 | 0.54 |
| This village puts me in a nostalgic mood | 3.77 | 0.68 | NA | 0.462 | | |
| This village induces feelings and sentiment | 4.18 | 0.747 | 14.394 | 0.559 | | |
| This village makes me respond in an emotional manner | 4.13 | 0.779 | 14.785 | 0.606 | | |
| Think | | | | | 0.85 | 0.58 |
| This village makes me think | 4.12 | 0.723 | 17.424 | 0.523 | | |
| This village appeals to my creative thinking | 3.89 | 0.715 | 17.168 | 0.511 | | |
| I engage in a lot of thinking in this village | 4.04 | 0.793 | 20 | 0.629 | | |
| This village stimulates my curiosity | 3.97 | 0.815 | NA | 0.663 | | |
| Act | | | | | 0.86 | 0.60 |
| I would like to take pictures in this village as mementos | 4.09 | 0.728 | NA | 0.531 | | |
| I would like to share experiences in this village | 4.06 | 0.847 | 18.571 | 0.717 | | |
| I would like to engage in physical activities | 4.04 | 0.831 | 17.789 | 0.691 | | |
| I would like to search related information | 3.87 | 0.688 | 14.606 | 0.474 | | |
| Relate | | | | | 0.85 | 0.59 |
| This village induces in me a sense of identity towards ecological conservation | 3.89 | 0.745 | 17.066 | 0.555 | | |
| This village reminds me of social arrangements | 3.89 | 0.818 | 19.104 | 0.669 | | |
| This village makes me think about relationships | 3.94 | 0.726 | 16.635 | 0.526 | | |
| I can relate to other people through visiting this village | 3.75 | 0.79 | NA | 0.624 | | |
| **Brand Image** | | | | | | |
| Functional | | | | | 0.85 | 0.65 |
| This village provides both educational and entertaining activities | 4.30 | 0.753 | NA | 0.567 | | |
| This village allows for stress relief | 4.19 | 0.832 | 18.341 | 0.693 | | |
| This village provides healthy leisure activities | 4.18 | 0.838 | 18.467 | 0.703 | | |
| Experiential | | | | | 0.86 | 0.61 |
| This village is a place with a variety of attractions | 4.17 | 0.794 | NA | 0.631 | | |
| This village is pleasant | 4.18 | 0.795 | 19.322 | 0.632 | | |
| This village is full of fun | 3.90 | 0.762 | 18.314 | 0.581 | | |
| The activities in this village impressed me a lot | 4.09 | 0.76 | 18.247 | 0.577 | | |
| Symbolic | | | | | 0.84 | 0.56 |
| This village is full of arts/humanity atmosphere | 4.06 | 0.811 | NA | 0.657 | | |
| This village has abundant drum-related culture | 4.30 | 0.75 | 17.998 | 0.562 | | |
| This village is rich in ecology | 3.90 | 0.757 | 18.234 | 0.574 | | |
| This village has cordial interpreters with expertise | 4.21 | 0.673 | 15.743 | 0.453 | | |

*4.2. Measurement Model*

Before investigating the hypothesized relationships between constructs, confirmatory factor analysis (CFA) was used to verify the proposed factor structure. A two-step approach was applied as suggested by Anderson and Gerbing [110] to validate the multi-attribute scales. The first step involved the validation of the measurement model for each construct, followed by the testing of the overall measurement model. For visitors' experiences, four items were deleted because of low standardized loadings and error covariance with another variable. Two items measuring brand image were removed due to low standardized loadings. The goodness-of-fit indices for the measurement of constructs are shown in Table 4 and indicated a satisfactory model fit with the data [115]. Composite reliability and

construct validity was evaluated next to validate the psychometric properties of measurement scales. As observed in Table 3, the reliability of the scales was confirmed because the composite reliability indices of each of the dimensions obtained were higher than 0.6 [116]. Construct validity was examined through convergent validity and discriminant validity. Convergent validity was demonstrated in two forms: all factor loadings being significant and greater than 0.5, and average variance extracted (AVE) for each of the factor being above 0.5, ref. [117] except for the sense dimension. Discriminant validity was assessed following Fornell and Larcker's [117] suggestion by comparing the square root of the AVE for each factor and correlations across the constructs. The results suggested sound discriminant validity as more than 75% of the comparisons achieved the criteria as suggested by Hair et al.; namely, the square root of the AVE should be greater than the factor inter-correlations.

**Table 4.** Goodness-of-fit indices for the measurement models.

| Construct | $\chi^2$ | *p* | CFI | NNFI | RMSEA |
|---|---|---|---|---|---|
| Experience | 349.03 | 0.000 | 0.956 | 0.946 | 0.06 |
| Brand image | 146.11 | 0.000 | 0.968 | 0.956 | 0.072 |
| Overall measurement model | 119.56 | 0.000 | 0.985 | 0.98 | 0.054 |

After the validation of the two key measurement scales, the final measurement model was specified, which included 4 latent variables and 12 observed variables. Visitor experiences and brand image were specified as a first-order model and reflected by factors based on the CFA results. The overall measurement model was tested using CFA and the results showed a satisfactory model fit (Table 4). Regarding the properties of scales, composite reliability was measured and the results demonstrated good internal consistency with values ranging from 0.89 to 0.91 (Table 5). Convergent validity was ratified by significant factor loadings (>0.5) and high average variance extracted (AVE > 0.5) as observed in Table 5. Discriminant validity was also confirmed by all correlations between constructs less than 0.82 (the least of the square root of AVE for each construct) (see Table 6). In summary, the assessment of the measurement model showed strong evidence of reliability and validity of the scale.

**Table 5.** Results of the overall measurement model.

| Constructors and Items | Factor Loadings | *t*-Value | Squared Multiple Correlation | Composite Reliability | AVE |
|---|---|---|---|---|---|
| Experience | | | | 0.91 | 0.67 |
| Sense | 0.781 | NA | 0.61 | | |
| Feel | 0.806 | 19.636 | 0.649 | | |
| Think | 0.884 | 22.103 | 0.782 | | |
| Act | 0.843 | 20.812 | 0.711 | | |
| Relate | 0.78 | 18.861 | 0.609 | | |
| Brand Image | | | | 0.89 | 0.73 |
| Functional | 0.774 | NA | 0.599 | | |
| Experiential | 0.863 | 21.171 | 0.745 | | |
| Symbolic | 0.921 | 22.863 | 0.848 | | |
| Satisfaction | | | | 0.80 | 0.80 |
| Satisfaction | 0.894 | NA | 0.80 | | |
| Loyalty | | | | 0.90 | 0.74 |
| Revisit | 0.857 | NA | 0.734 | | |
| Recommend | 0.847 | 23.656 | 0.717 | | |
| Share | 0.878 | 25.018 | 0.771 | | |

**Table 6.** Discriminant validity of the scale.

|  | 1 | 2 | 3 | 4 |
|---|---|---|---|---|
| 1. Experience | 0.82 |  |  |  |
| 2. Brand Image | 0.78 | 0.85 |  |  |
| 3. Satisfaction | 0.61 | 0.56 | 0.89 |  |
| 4. Loyalty | 0.68 | 0.76 | 0.55 | 0.86 |

Note: correlation estimated between the factors are presented below the diagonal line. numbers in the diagonal line are square root of average variance extracted between the factors.

### 4.3. Structural Model

Given sound psychometric properties for each factor across the four constructs, we next performed an analysis of the proposed causal relationships between constructs. The results of the structural model are presented in Table 7 and showed an overall good fit: $\chi^2 = 119.566$, df = 49, $p < 0.001$, CFI = 0.985, NNFI = 0.98, RMSEA = 0.054. The path analysis results provide support for five of the first six hypotheses, since all the estimates among these constructs were positive and statistically significant at the 0.05 level. Visitors' experiences showed significant effects on the brand image ($\beta = 0.87$, $p < 0.001$) and overall satisfaction ($\beta = 0.53$, $p < 0.001$); hence hypothesis 1 and hypothesis 2 are supported. The brand image positively affected overall satisfaction ($\beta = 0.20$, $p < 0.05$) and loyalty ($\beta = 0.79$, $p < 0.001$); hence hypothesis 4 and hypothesis 5 are supported. The sixth hypothesis is also supported since the effect of overall satisfaction on loyalty was significant at the 0.05 level ($\beta = 0.15$, $p < 0.05$). However, the direct effect of visitors' experiences on loyalty was statistically nonsignificant ($\beta = -0.044$, $p > 0.05$); hence hypothesis 3 is rejected. The modified structural model is shown in Figure 2.

**Table 7.** Results of the proposed model.

| Hypothesized Path | $\beta$ | $t$ Value | $R^2$ | Hypothesis Testing |
|---|---|---|---|---|
| H1: Experience → Brand Image | 0.87 | 16.43 *** | 0.75 | Supported |
| H2. Experience → Overall Satisfaction | 0.53 | 5.09 *** |  | Supported |
| H3: Experience → Loyalty | −0.044 | −0.508 |  | Not Supported |
| H4: Brand Image → Overall Satisfaction | 0.20 | 1.98 * | 0.51 | Supported |
| H5: Brand Image → Loyalty | 0.79 | 9.03 *** | 0.74 | Supported |
| H6: Overall Satisfaction → Loyalty | 0.15 | 2.74 ** |  | Supported |
| H7: Experience → Brand Image → Loyalty | NA |  |  | Supported |
| H8: Experience → Overall Satisfaction → Loyalty | NA |  |  | Supported |

\* $p < 0.05$, \*\* $p < 0.01$, \*\*\* $p < 0.001$.

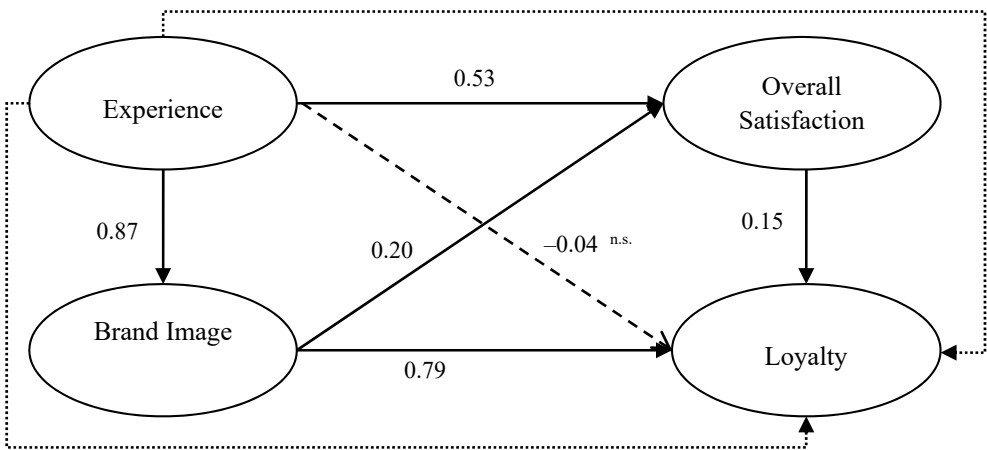

**Figure 2.** The modified structural model.

We further analyzed the indirect effects through the bootstrapping method in order to examine if brand image and overall satisfaction were significant mediators of the relationship between visitor experiences and loyalty. The results are reported in Table 8 and showed a total indirect effect of visitors' experiences on loyalty as 1.088. The result affirmed the existence of mediation effects since its 95% CI does not contain zero and also indicated that visitors' experiences affected loyalty only through mediators, because its direct effect was not significant and less than the total effect. Besides, PRODCLIN2 by MacKinnon et al. [113] was used to determine the existence of specific indirect effects. In the current study, three specific indirect effects are shown in Figure 1 and proved to be significant in that no zero was contained in the 95% CI (Table 8). The seventh hypothesis (visitors' brand image mediates the effects of experience on their loyalty towards the destination) and the eighth hypothesis (visitors' overall satisfaction mediates the effects of experience on their loyalty towards the destination) are therefore supported in that both visitors' brand image and overall satisfaction had mediating effects on the relationship between experience and loyalty. Lastly, to determine the relative importance of these specific indirect effects, the percentage of each specific indirect effect that accounted for the total indirect effect was calculated. The specific indirect effect is defined as the product of the two unstandardized paths linking X to Y via that mediator and is as follows: EXP → BI → LOY (0.793 × 1.193 = 0.946), EXP → SAT → LOY (1.481 × 0.072 = 0.107), EXP → BI → SAT → LOY (0.793 × 0.624 × 0.072 = 0.036), respectively. As a result, the most important indirect effect was through brand image to influence loyalty, which accounted for 86.95% (0.946/1.088) of total indirect effect, followed by through satisfaction (0.107/1.088 = 9.83%), and through both brand image and satisfaction (0.036/1.088 = 3.31%).

**Table 8.** Mediation effects of visitors' brand image and overall satisfaction.

| Variables | Point Estimate | Bootstrapping | | | | MacKinnon PRODCLIN2 95% CI | |
| | | Bias-Corrected 95% CI | | Percentile 95% CI | | | |
| | | Lower | Upper | Lower | Upper | Lower | Upper |
|---|---|---|---|---|---|---|---|
| Total Effects | | | | | | | |
| EXP → LOY | 1.027 | 0.906 | 1.161 | 0.905 | 1.161 | | |
| Total Indirect Effects | | | | | | | |
| EXP → LOY | 1.088 | 0.824 | 1.381 | 0.822 | 1.374 | 0.67 | 1.27 |
| Specific Indirect Effects | | | | | | | |
| EXP → BI → LOY | 0.946 | | | | | 0.67 | 1.26 |
| EXP → SAT → LOY | 0.107 | | | | | 0.02 | 0.24 |
| EXP → BI → SAT → LOY | 0.036 | | | | | 0.01 | 1.06 |
| Total Direct Effects | | | | | | | |
| EXP → LOY | −0.06 | −0.335 | 0.203 | −0.334 | 0.204 | | |

Note: EXP: visitor experiences; LOY: loyalty; BI: brand image; SAT: satisfaction.

## 5. Discussion

Drawing on the experiential marketing construct, we examined tourists' experiences at a sugar heritage destination and the causal relationships between experience, brand image, satisfaction, and loyalty. Although the marketing literature has studied the effect of experience on satisfaction and loyalty, insufficient work has been done on visitors' experiences in industrial heritage tourism, and specifically on how the visitors' experiences at a sugar heritage destination affect satisfaction and loyalty. This study also includes the mediation effect of brand image and satisfaction on the relationship of experience to loyalty. Our results demonstrated that visitors' experiences at a sugar heritage destination were reflected in five experiential modules as proposed by Schmitt [69]. As for the hypothesis testing, overall, our findings support the contention that brand image and satisfaction are significant predictors of loyalty, with brand image being a stronger predictor than

satisfaction. However, visitors' experience had no direct effect on loyalty. Brand image and satisfaction fully mediated the relationship between visitors' experiences and loyalty.

The results contribute to the existing body of knowledge in two ways. First, the study adds to the heritage tourism literature by demonstrating that the application of the experiential marketing construct in visitors' experiences at a sugar heritage destination is confirmed by the emerging five dimensions: sense, feel, think, act, and relate. This factor structure is consistent with Schmitt's [69] claim and Tsaur, Chiu, and Wang's [118] as well as Chang's [103] results. Our results also suggest that visitors' experiences are most driven by how they feel, think, and act as they encounter rich stimuli from the destination. For example, the specialized interpretation of the sugar refining history and drum performers' training process provided by the village stimulates visitors' curiosity and encourages thinking. The drum teaching course allows visitors to engage physically in the drum beating activities, while the relaxing landscaping, renovated stage design, and well-maintained sugar refining equipment induce sentiments and emotions. In our study, the sense experience, claimed to be the easier trigger, was not as significant as in other cases [69,118]. This difference coincides with the aims of Ten Drum Culture Village: promote drum music and preserve the history of the sugar industry. In an attempt to achieve those objectives, the village is focused more on visitors' affective and intellectual responses instead of sensory stimulation, and its efforts appeared to be successful in terms of visitors' perception.

Second, this study advances our understanding about the determinants of loyalty, their relative influence, and the mediating effect of brand image and satisfaction on the relationship between experiences and loyalty in tourism literature. Concerning the roles of experience, brand image, and satisfaction in predicting loyalty, brand image is a more powerful determinant of loyalty than satisfaction based on the standardized path coefficients. In stimulating visitors' loyalty behaviors, the favorable image that a visitor has towards the destination is more significant than perceived experience and satisfaction. This result is in line with the findings of Andreassen and Lindestad [119], who reported the corporate image, not satisfaction, significantly affects loyalty, and corresponds to the argument that brand image and affect (experiences) must be combined into the model to best predict loyalty [120]. However, some researchers reported a strong relationship between satisfaction and loyalty. A plausible explanation for the mixed findings is that the study context differs from that of previous research. For example, Kozak and Rimmington [98] investigated visitors to Mallorca, a famous destination among British citizens. In Yoon and Uysal's [121] research, Northern Cyprus was chosen as the study site, known for its archeology and sandy beaches. In this study, we chose a sugar heritage destination rich in industrial heritage and drum culture, which is more experiential-oriented, and attracts those visitors whose needs are quite different from vacationers', as investigated by other researchers. Hence the benefits-based brand image perceived by visitors is important in assisting them in making decisions about future behaviors. This is also reflected in visitors' evaluation on brand image, with functional image (i.e., educational and entertaining activities, relaxing environment, healthy leisure activities) being rated the highest, followed by symbolic and experiential images.

Visitors' experiences did not directly influence their loyalty towards the destination as opposed to the argument by Schmitt [69] and Barnes et al. [73]. This is not surprising, as experience has been reported to affect loyalty behaviors through emotion and value perception [72,122]. In other words, the effect of the stimulus outcome (experience) on consumer behavior is mediated by the emotional state. However, experience did directly affect brand image and accounted for more than 75% of variance, which complements the theoretical assumptions that brand image perceived by visitors is created as the experiential outcome of their exposure to stimuli by providing empirical support to the relationship. This is also consistent with previous findings, suggesting that experiences perceived directly influenced the associations linked to a specific brand and ultimately resulted in a favorable positive brand image [123,124].

By conducting mediation analysis, our results also illustrate the key mediating role of brand image on the relationship between visitors' experiences and their loyalty towards the destination. Among the three proposed indirect effect routes, the majority of the mediational effect is in fact occurring through brand image, with 86.95% of the effect being mediated by brand image. Around 10% of effect is attributed to the mediation of satisfaction on loyalty. That is, experience does not have direct influence on loyalty, though it is mediated through brand image to influence loyalty. Visitors who considered the visits to this village touched their hearts, made them think, and engaged in action tended to have a more favorable and positive brand image of the village and that led to a higher level of loyalty behaviors. However, this relationship contradicts some study results and needs further examination.

## 6. Conclusions

The purpose of this study was to explore the induced tourist experiences after exposure to marketing stimuli during visits to a sugar heritage destination, and to attempt to extend the theoretical and empirical evidence about the causal relationships between experience, brand image, satisfaction, and loyalty. We consider this context, a sugar heritage destination, to be a relevant contribution, given that previous works analyzing experience and satisfaction on loyalty have focused on mass travel attractions and less research included the construct of brand image in the modeling of loyalty. This study advances the current knowledge of the relationships between experience, brand image, satisfaction, and loyalty by examining the direct and indirect (mediation) effects on loyalty. Overall, our model of loyalty was confirmed due to its high validity and explanatory power. The study results empirically support the application of experiential marketing in a sugar heritage destination and suggest that visitors' experiences are most driven by how they feel, think, and act. Visitors' experiences induce the brand image, and in turn, brand image influences satisfaction and loyalty.

The study highlights the importance of brand image on loyalty formation and its mediating role between experience and loyalty. The results advance the findings from previous research indicating that visitors' perceived experience directly influences the associations linked to a specific brand that are helpful in forming a favorable brand image, and ultimately lead to a higher level of loyalty. In addition, the synergetic effects of experiences and brand image simultaneously contributing to visitor satisfaction are confirmed and correspond to previous research statements. While comparing the influential paths of brand image and experience to satisfaction, we can affirm that the effect of experience on satisfaction is greater than the effect of brand image. On these grounds, experiential marketing plays a critical role in the evaluation of brand image and satisfaction. The study makes a contribution to the field of heritage tourism by proposing and examining a conceptual model of loyalty from experiential marketing perspective, which was not explored explicitly in prior research.

### 6.1. Managerial Implications

The results derived from this study can provide tourism practitioners with insights into loyalty-building endeavors. Through unveiling the mediating effects of brand image and satisfaction on loyalty, this study shed lights on the need to create desired visitor experiences as positive visitor experiences affect the brand image, which further leads to preferred loyalty. This can be done by implementing the experiential marketing approach, in particular, providing opportunities to encourage visitors to feel, think, and act; for example, by creating atmosphere which is attached to visitors' childhood memory or happy/relaxing emotions, by landscaping and decorating the village with vintage collections in creative ways. Even when some visitors do not have direct access to the past, they can develop nostalgic feelings from the settings and share with their families. Marketers could also provide visitors with diverse on-site recreational experiences, and this could be in forms of adventure recreation facilities, interactive interpretation media, special events,

heritage eating experiences, and souvenir sales. Given that the village was once a sugar refinery, an experiential course featuring brown sugar processing could be provided. That course not only allows visitors to establish links to their life but encourages them to show more interest in the past of the sugar industry, and achieves the objective of promoting the identity of the local drum music group. Moreover, to satisfy visitors' need of souvenirs and enhance their memory of the trip to this village, stylish sugar-related products could be created for the village at an affordable price. As noted by Lane et al. [14], industrial heritage visitors' experiences are the best when the tourism is accompanied by other activities such as events, sports, theater, and shopping possibilities.

Furthermore, certainly of importance to marketers would be the perceptions (image) these visitors possess towards the village. The image should fully reflect the unique attributes that differentiate this village from its competitors. The marketers should focus their efforts more strongly on managing positive image by using an integrated marketing communication strategy. Through a combination of promotional tools, the village could clearly convey consistent and complementary messages to attract visitors and enhance their revisit experiences. For example, the village could take advantage of the social media trends and co-creation approach by encouraging visitors to take pictures or make videos on specific topics and post on the designated media. Rewards could be offered such as free entrance or limited souvenirs for those who convey the village's educational and drum-related culture in their work in the most attractive style. Through this type of campaign, the village could develop ties with visitors and manage customer relationships on social media via a fan page. Using a fan page as a platform for information sharing, events/activity promotion, and technical/service support could help enhance visitors' impressions towards the village, which in turn, will increase their loyalty. Moreover, a more vigorous public relations strategy is recommended. The village needs to actively provide promotional materials to traditional media, by demonstrating its efforts on the issues of sustainability, culture, and heritage conservation, drum music education, and personnel training. Together with those approaches described above, the village could enhance the holistic visitor experience and create a positive and favorable image.

### 6.2. Limitations and Future Research

Despite the contribution of this study, it is subject to the following limitations. First, the model tested in this study is only for actual visitors and the difference between new and repeat visitors remained outside this research. As noted by Kotler and Armstrong [76], the formation of image differs depending on personal experiences. There is a need to compare the perceived brand image of new versus repeat visitors, in order to learn the structure of brand image formation. Second, the construct of brand image originated from the research of products, and hence the application of its measurement to destinations needs to be further tested given the complex nature of destinations, although there are increasing studies on destination branding. Third, with one sugar heritage destination, the findings may not be generalized to various types of heritage destinations. To increase the generalizability of the findings for heritage destinations, we recommend testing the model in other segments (e.g., heritage destinations, ecotourism destinations, culinary destinations). A comparison across multiple destinations is also recommended to better understand the experience structure and the mediating effects of brand image and satisfaction.

**Author Contributions:** Conceptualization, A.H.C. and R.Y.W.; writing—original draft preparation, A.H.C.; data collection, A.H.C.; software and data analysis, A.H.C.; validation, A.H.C. and R.Y.W.; visualization, A.H.C.; writing—review and editing, A.H.C. and R.Y.W. All authors have read and agreed to the published version of the manuscript.

**Funding:** This research received no external funding.

**Institutional Review Board Statement:** Not applicable.

**Informed Consent Statement:** Informed consent was obtained from all subjects involved in the study.

**Data Availability Statement:** The data are available on request from the corresponding author.

**Acknowledgments:** We thank all parties that helped with the data collection process.

**Conflicts of Interest:** The authors declare no conflict of interest.

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
