# Peer review of "Mediating Effect of Brand Image and Satisfaction on Loyalty through Experiential Marketing: A Case Study of a Sugar Heritage Destination"

_sustainability, doi:10.3390/su14127122_

Round 1

Reviewer 1 Report

I enjoyed reading this paper. 

The paper presents an interesting topic.  I think this is a good paper, very clear and, methodologically substantial.  

Author Response

Thank you, we greatly appreciate your acknowledgement of our work.

Reviewer 2 Report

This study proposes a structural model to investigate the relationships between experience, brand image, overall satisfaction, and tourist’s loyalty to a sugar heritage destination. The findings may help the destination managers devise marketing strategies to encourage their tourists to visit continuously. Despite using a fairly sophisticated statistical testing, the study has a number of fundamental flaws in the overall structure, data analyses, and discussion.

  1. In the Introduction section, the author(s) covered previous studies on determining tourists’ loyalty and behaviors. Several factors influence heritage tourists to visit and revisit a destination. The following remains ambiguous: in this study, why are experience, brand image, overall satisfaction considered to be the most important variables in determining heritage tourists’ loyalty to a heritage destination?
  2. It would be necessary for the author(s) to provide a reasonable explanation and justify why the role of brand image and satisfaction as mediating variables in the relationship between experience and loyalty should be examined.
  3. Related with the same issue as above, what is the theory-based connection between the objective of this study (i.e., a proposed model) and the heritage tourists’ loyalty/behaviors within the context of a heritage destination?
  4. The importance of this study seems weak or poorly elucidated. What is the importance of this study?
  5. The Literature Review section is weak and insufficient. What is the theory-based connection between the objective of this study and tourists’ behaviors within the context of a heritage destination?
  6. There was no detailed description of and for the data collection, for example, the composition of the population of this study and the sampling method used. Where were the participants interviewed?
  7. The discussion is generally unremarkable, lacking in data-based discussion and conclusion. Although this study confirmed that loyalty was formed by the interrelationships among study constructs, no discussion regarding how the results contribute significantly to hospitality literature is provided.
  8. What is the theoretical contribution of this study to the tourist behavior literature in general and loyalty in particular?

Author Response

Thank you for your comments and suggestions, and we have revised our paper accordingly. Please see the attachment.

Reviewer 3 Report

The study is comprehensive. The high number of hypothesis (8) might be confusing in the begining but they were successfully explained and analised further in the paper. Referencing system is adequate. The research methodology used is sound and clear described (SPSS for descriptive and item analysis,structural equation models AMOS v.24 in the model  and hypothesis testing). Results are commented in detail.

Author Response

Thank you very much, we greatly appreciate your acknowledgement of our work.

Reviewer 4 Report

Dear Authors,

I recommend your paper entitled: "Mediating effect of brand image and satisfaction on loyalty through experiential marketing: A case of sugar heritage destination" for publication in Sustainability in its current form.

Best regards

The reviewer

Author Response

(The authors gave the same response as above.)

Round 2

Reviewer 2 Report

The paper has been improved significantly and is fine in all major respects. I recommend that this paper can be accepted for publication.